# Threats to the Dignity of People with Advanced Illness Who Are Treated in Emergency Departments: A Qualitative Study

**DOI:** 10.3390/healthcare12242581

**Published:** 2024-12-22

**Authors:** Alba Fernández-Férez, Ousmane Berthe-Kone, Gonzalo Granero-Heredia, Matías Correa-Casado, María del Mar Jiménez-Lasserrotte, Álvaro Martínez-Bordajandi, José Granero-Molina

**Affiliations:** 1International Doctoral School, University of Almeria, 04120 Almeria, Spain; alba.fernandez.ferez.sspa@juntadeandalucia.es (A.F.-F.); ggh736@inlumine.ual.es (G.G.-H.); 2Servicio Andaluz de Salud, 04005 Almeria, Spain; ob053@inlumine.ual.es; 3Department of Nursing, Physiotherapy and Medicine, University of Almeria, 04120 Almeria, Spain; mjl095@ual.es (M.d.M.J.-L.); jgranero@ual.es (J.G.-M.); 4Anaesthesiology and Surgical Critical Care Department, Torrecardenas University Hospital, 04009 Almeria, Spain; amb943@inlumine.ual.es; 5Facultad de Ciencias de la Salud, Universidad Autónoma de Chile, Santiago 7500000, Chile

**Keywords:** advanced illness, dignity, emergency department, end-of-life care, threats

## Abstract

Background: Dignity is a key element in end-of-life care. Patients with advanced illness attend the emergency department to seek symptom relief but may find their dignity under threat in these services. Objective: The purpose of the study was to explore the threats to dignity perceived by people with advanced illnesses who are treated in emergency departments. Methods: A descriptive qualitative study was designed for which 18 patients with advanced illnesses were interviewed after being seen in an emergency department. The study complied with the principles of the Declaration of Helsinki for medical research involving human subjects. Results: Two themes were developed from the data analysis that shed light on how patients with advanced illnesses perceive threats to their dignity when seen in emergency departments: (1) “when care focused on diagnosis and treatment limits the dignity of the patient with advanced illness” and (2) “the social dimension of dignity in people with advanced illness in emergency departments”. Conclusions: We conclude that the structural, organizational, and care characteristics of emergency departments may pose a threat to the dignity of people with advanced illnesses who attend the emergency department. Family members, professionals, and other patients can both guarantee and threaten the dignity of people with advanced illnesses when they are treated in emergency departments.

## 1. Introduction

Human dignity is a multidimensional concept that is associated with a value inherent in every person regardless of age, gender, intellectual abilities, health status, socioeconomic status, religion, or nationality [1]. Dignity is at the core of the UN Universal Declaration of Human Rights [2]. From a Kantian perspective, dignity is rooted in the autonomy and free will of rational beings, who are capable of self-legislation and must treat themselves and others as ends in themselves, never merely as means to other ends. Dignity represents an intrinsic, immeasurable value derived from a person’s inner worth [3]. In recent decades, the concept of human dignity has become particularly relevant in bioethics, notably the relationship between dignity and end of life [4]. Dignity is associated with a person’s autonomy, as well as their right to make decisions about their own life [5]. This includes decisions about medical treatment and care that ensure a dignified death for people with an advanced illness [6].

An advanced, terminal illness is an incurable, advanced, and progressive disease with a limited prognosis for life and little chance of responding to specific treatments [7]. It is estimated that more than 40 million people need end-of-life care globally, a figure that could double in a few years [8]. Some studies estimate that in Spain, 1% of the adult population suffers from an advanced illness [9]. Advanced illness varies in nature, causing physical, physiological [10], psychological, emotional, and spiritual [11] changes in the person affected, including pain, dyspnea, anxiety [12], and loss of dignity [13]. This has a significant impact on individuals and their families [14,15], which, in turn, leads to a high demand and use of resources to receive the care required [16]. In this regard, emergency departments (EDs) are among the most in-demand services [17,18]; an estimated 70% of people with advanced illness attend an ED at least once during their last 6 months of life [19,20]. In these highly technical services, care is based more on saving lives [21] than on viewing the patient holistically [22], which may result in the individual’s dignity being undermined [22,23]. The perception of being treated with a lack of dignity can lead to feelings of vulnerability [24], depersonalization [25], and an increased longing for death [26].

H.M. Chochinov’s model of dignity preservation [27] established three categories that define the phenomenon: (1) illness-related concerns, (2) dignity-conserving repertoire, and (3) social dignity inventory. This model was developed with terminally ill cancer patients receiving care at home. There is also a wealth of research on dignity in EDs from the point of view of professionals [28,29] and family members [13], as well as patients in the hospital setting [30]. However, there is a lack of research on the threats to dignity perceived by people with advanced illnesses who attend EDs [31]. Aligning research and policy with patients’ priorities and perceptions would help to ensure the value of research for society [18]. The aim of this study is to explore the threats to dignity perceived by people with advanced illnesses who are treated in emergency departments.

## 2. Materials and Methods

### 2.1. Design

A descriptive qualitative study design was conducted. This qualitative design is appropriate when health science researchers seek to share individual narratives, understand the context or setting of problems, develop theories, and when traditional quantitative statistical analyses do not address the problem at hand [32]. In writing the report, the recommendations of the COREQ guide were followed [33].

### 2.2. Participants and Context

Patients diagnosed with advanced illnesses who had been treated in one of the three public hospitals in the provinces of Almeria and Granada in southern Spain were selected for the study. Inclusion criteria included (1) being patients with advanced disease and (2) having received care in the ED at least once in the last 6 months. Exclusion criteria included (1) being a minor, (2) having cognitive impairment, and (3) having a condition that prevented them from maintaining a conversation. For the recruitment of the participants, one of the members of the research team contacted the lead nurse in the ED of each hospital, who provided the contact details of the patients who met the inclusion criteria. The case management nurse was then contacted to arrange an appointment with the patient, whether they were admitted to the hospital or at home. Twenty-three patients were invited, of whom 18 participated and 5 declined due to their clinical situation. The sociodemographic data of the participants are given in Table 1.

### 2.3. Data Collection

The interviews took place between 2020 and 2023. Individual in-depth interviews were conducted in the hospital ward or at the patient’s home. They were led by nurses who had not previously provided care to the patient in the ED. Hence, an interview protocol was designed beforehand (Table 2). The interviewers had received training in qualitative research (master’s degree) and had tested the interview script. The interviews lasted an average of 35 min, as the patient’s clinical situation did not allow for more time. The interviews were recorded using voice recorders to avoid bias and to facilitate the analysis of the information. The audio files were saved for later transcription. Due to the clinical condition of the patient, the content of the transcripts or the analysis was not shown to them for their approval or to provide further information. After analyzing the first interviews, they continued until no new themes emerged, and the researchers considered that data saturation had been reached.

### 2.4. Data Analysis

All recordings were transcribed for analysis. The transcripts were entered into an ATLAS.ti project, together with notes made by the researchers. For the thematic analysis, an inductive strategy was used following the phases described by Braun and Clark [34].

Phase 1. Familiarization with the data, which consisted of a complete reading of all the transcripts to gain a general understanding of the content, followed by a re-reading, in which annotations were written using the ATLAS.ti memo function.Phase 2. Systematic data coding, in which the transcripts were coded using various procedures of the ATLAS.ti 24 software. Codes with a similar meaning were grouped into units of meaning.Phase 3. Generate initial themes from codes and collated data. Units of meaning were grouped into themes representing shared patterns of meaning, linked by a central concept or idea.Phase 4. Develop and review themes, checking that the themes were consistent with the codes they grouped together and with the quotations coded with these codes. Networks were then created in ATLAS.ti (Figure 1).Phase 5. Defining and naming the themes, refining the analysis and naming of each theme.Phase 6. Writing the report. During writing, the most representative examples (quotations) were selected, summaries of meaningful examples were made, and the analysis was related to the research question and the literature.

An example of the coding and categorization process with ATLAS.ti is shown in Figure 1.

### 2.5. Ethical Issues

All participants were informed of the objectives and methodology of the research, as well as their right not to answer questions or to withdraw from the study at any time during the research without any repercussions. They gave their informed consent prior to data collection. The project was approved by the Almeria Research Ethics Committee, FFI2016-76927-P (AEI/FEDER, EU). Data processing was carried out in accordance with the Personal Data Protection Act [35]. The data were stored in a protected folder to which only the researchers had access. Pseudonyms were given to the participants.

### 2.6. Rigor

The criteria to assess the rigor of qualitative research in emergency medicine (credibility, dependability, confirmability, and transferability) were used [36]. Credibility was sought through peer debriefing, negative case analysis, and member checks. Dependability was sought through a detailed description of processes and procedures, so that they could be audited. To ensure confirmability, several researchers checked that the data (participants’ quotations) were consistent with the codes, themes, and subthemes developed. Transferability was ensured by thick description, purposive sampling, and reflexivity of the researchers.

## 3. Results

Eighteen patients participated in this study, of whom 66.7% were male and 33.3% were female. The age range of the participants was between 51 and 88 years (mean 71.8, SD: 10.5). Two main themes were developed from the data analysis: (1) individual strategies for the preservation of dignity: thinking about life and deciding about death and (2) the social dimension of preserving dignity. The themes and subthemes allow for an exploration of the dignity-preserving strategies of patients with advanced illnesses who are treated in EDs (Table 3).

### 3.1. When Care Focused on Diagnosis and Treatment Limits the Dignity of the Patient with Advanced Illness

This theme highlights the characteristics of ED care and how these contribute to the loss of dignity for people with advanced and/or terminal illnesses. The first subtheme reflects how the care received is focused on identifying and solving problems, while the second describes how structural and organizational characteristics of EDs have an impact on patient dignity.

#### 3.1.1. Dignity and Vulnerability of People with Advanced Illness in Need of Emergency Care

This subtheme refers to the health-related reasons for seeking emergency care and how patient perception of dignity changes. ED care is centered on reversing acute, life-threatening, or extremely serious situations. This study focuses on cases of advanced or terminal illness, which involve specific circumstances that are not always taken into account. Participants feel particularly vulnerable in a situation where they perceive that they are losing control over their own lives.


*They don’t kill us, but when you go in as a patient you know they can do whatever they want to you… you feel absolutely helpless. *
(P. 12)

Triage systems in EDs are generally focused on alleviating critical, acute, life-threatening situations, as opposed to advanced, chronic, or terminal illnesses. This can lead patients to feel that they are second-class patients (not a priority), despite the gravity of their situation. This inevitably results in a perceived loss of dignity.


*When you see that they give preference to an accident or a heart attack and they make it clear that oncology patients do not have preference when it comes to care… you think: Is mine not serious? They even tell you that since I had cancer, why did I want it (care) if I was going to die? *
(P. 6)

The patients’ vulnerability stems from the advanced, sometimes terminal, nature of their illness. Various driving forces lead them to attend the ED, including their unstable condition, crisis moments, or family members’ fears of adverse events occurring at home. The participants perceive that in the ED, they are often not provided with care that goes beyond merely alleviating symptoms. This lack of concern for their complex and vulnerable situation is understood as a loss of dignity.


*The impression is that you go, they give you a bottle of Nolotil, they relieve the pain, and you go home. And for me, I don’t feel right in those situations. It’s like… you wonder, why did I come here?*
(P. 7)

The participants perceive that professionals make an effort to establish a diagnosis, for which they need to gather information through observation, examination, complementary tests, etc. However, they often feel that this effort is misdirected and irrelevant and disregards their need for comfort and reassurance.


*I’m dying, so why tell me that you don’t know what’s wrong? What are you going to explore me for? What are you going to give me in exchange for knowing what I already have? [insult]! So give me some comfort, put me in a bed, put me to sleep, let me be calm, let me not be in pain, and that’s what you have to provide.*
(P. 18)

#### 3.1.2. Structural and Organizational Constraints of EDs and Their Impact on Dignity

The subtheme reflects patient perceptions of the structure and organization of the ED, which is evidently linked to patient dignity. For example, waiting time is viewed more positively simply when a comfortable seat is available.


*To be there for 12 to 14 h… I would have needed something softer on top of the stretcher. Something more comfortable.*
(P. 3)

Long waiting times for care are perceived as degrading. The wait is sometimes twice as long if a regional hospital does not have the necessary means or resources, and the patient needs to be transferred to a main hospital, where the process (admission, triage, observation) is repeated.


*The first time I was hospitalised in Huércal, I spent 12 h in observation in Huércal Overa. But then from Huércal Overa, they sent me to observation in Almería. But then they keep me there for another 12 h in observation, in Almería. Is that degrading or not? That is degrading.*
(P. 16)

The demand is sometimes so high that there is a lack of resources to accommodate all the patients, leading them to feel uncomfortable and neglected. The participants perceived that these conditions do not contribute to the dignity of the person being cared for or the situation itself.


*They kept me on the chairs until there were beds. I was already dizzy, my back hurt, everything hurt. The emergency department that day was so overloaded [with work/patients]. There were no beds anywhere. Those are not dignified conditions.*
(P. 4)

Moreover, if a patient is not given the priority, they expect to receive for their clinical condition and its respective symptoms, such as dyspnea, this can lead to feelings of anxiety, neglect, and helplessness.


*I got there with some shortness of breath and nothing, no one would attend to me… In the end, the waiting time was too long. I even became anxious and tachycardic … it made me very ill. It is a very upsetting situation because you find yourself in a situation of helplessness and no one is there to attend to you.*
(P. 17)

When workloads are high, patient intimacy is often a low priority. Professionals focus on getting their work performed with no time to think about the privacy of the person receiving care.


*They sometimes put curtains up, but at other times, in certain places, they don’t cover the patients and it’s obvious. There is no privacy because they strip you naked right there.*
(P. 14)

Due to the characteristics of EDs, hospitals have decided to limit visits in order to avoid overcrowding. In patients with extensive family support, the family must decide which of the family members should stay with the patient. The others are sometimes asked to leave the ED and waiting areas.


*It’s sad enough that you get sick, that your children come, that your family comes, and that a ‘they can’t be here’ comes too.*
(P. 17)

### 3.2. The Social Dimension of Dignity in People with Advanced Illness in Emergency Departments

The participants identified threats to patient dignity in EDs, which stem from the social context and interpersonal relationships. They understand that, just as dignity is a personal attribute that can be threatened by environmental conditions, there is also a social dimension. The social context can either heighten or undermine patient dignity. The participants highlighted that family members, other patients, and healthcare professionals can undermine the dignity of terminally ill patients in EDs.

#### 3.2.1. Healthcare Professionals as Both Protectors and Threats to Dignity

Some patients who attend the ED value healthcare professionals positively when they provide comprehensive, humane, and quality care. Therefore, the dignity of these patients depends to a large extent on the way in which the healthcare professionals treat them. Some of the participants made particular reference to nurses being “the backbone of the healthcare system”.


*They gave me a lot of love there. They’re great, they’re all great. I think they’re so nice. They have such a big heart… they are very humble people and for me that’s really important.*
(P. 1)

In contrast, many of the participants have had negative experiences due to the professionals’ lack of tact or sensitivity. According to the participants, this included professionals not showing up, not dedicating enough time to the patient, or lacking teamwork skills. In this regard, the participants perceived that dignity is undermined by insensitive or unempathetic treatment.


*Once they came to wash me, and I was crying because I was in so much pain and they told me ‘You have to suck it up…’.*
(P. 14)

It can sometimes be a struggle for patients to maintain their dignity while being treated in the ED. The participants call for a change in the way that healthcare professionals approach care provision. They believe that care should not focus exclusively on saving the patient’s life but also on alleviating their symptoms. The patients attend the ED to put an end to their suffering and to go home as soon as possible.


*So I tell nurses and doctors to change their mentality. They can sometimes help us just by calming the pain and getting us out of there (from the ED).*
(P. 8)

The participants highlighted the responsibility emergency health professionals have to act as guarantors of patient dignity through active listening and empathetic care. It is crucial to listen to patients and their life stories for them to feel valued and respected, which is an essential component of dignity. This not only improves the patient’s experience but is also fundamental to preserving their dignity in times of vulnerability. One of the participants explained this sentiment in the following way:


*They have to help you, give you time to tell your story. I know there isn’t much time (in the ED), but they have to take a minute out of their time to try to listen to you and give you their full attention.*
(P. 13)

However, many of the participants justify the healthcare professionals’ attitude on the understanding that they are also human and can make mistakes or have “bad days”. The psychological burden to which they are subjected, especially in an emergency situation, makes them vulnerable.


*They do what they think is best, because they are not creatures that have come from another world.*
(P. 8)


*We don’t always have a good day, and maybe it’s on that day I don’t get the right level of care.*
(P. 4)

Nonetheless, most of the participants perceived that ED healthcare professionals do not show interest in their feelings or emotions. Rather than treating the patients holistically, they are only concerned with solving clinical problems. For this reason, some of the participants define themselves as just another number in the ED.


*Doctors in the ED know their stuff, but sometimes they won’t even look you in the face. To them you are just Patient X. They don’t even know what I look like.*
(P. 9)

#### 3.2.2. How Patients, Family, and Loved Ones Shape Experiences of Dignity Preservation or Loss

The social context that shapes the experiences of dignity and loss of dignity among patients with advanced illnesses who attend EDs is multifactorial. It includes healthcare professionals, resources, and infrastructure, as well as the patients themselves, their families, and loved ones. One of the main factors that contribute to patient’s perception that their dignity is under threat is the feeling that they are being a nuisance or a burden to their loved ones.


*Every time I go to the ED, my children have to take time off work, they have to leave their families… and be here for hours, sometimes days. I don’t like to see that I’m a hindrance to their daily life.*
(P. 18)

To avoid this feeling of being a burden, they sometimes hide their own suffering or avoid talking about death with loved ones. They do not want to be pitied, which prevents them from sharing their suffering with loved ones, thus undermining their dignity. Furthermore, if patients do not communicate their discomfort effectively in the ED, healthcare professionals may underestimate the severity of their condition, which could thus affect the quality of care they receive.


*I mean, if I can endure the pain, why am I going to make others suffer?)*
(P. 10)


*I don’t want to talk to my son about my sorrows because we all have a lot of them. What am I going to tell him? That I’m sick? That I’m bored? That I can’t do what I want to do? I don’t want to worry them because they know what I have and what the situation is.*
(P. 15)

In EDs, healthcare professionals are often faced with difficult decisions about the treatment of terminally ill patients. Conflict can arise when family members are determined to preserve their loved one’s life at all costs rather than accepting that it is time to redirect the focus of care from treatment (that can no longer be curative) to comfort in order to prevent suffering. One of the participants expressed their perceptions in the following way:


*My sister R., who is 50 years old, … was tortured (in the ED). She was married to a very conservative Italian who insisted on having everything done to her (subjecting her to futile treatment in the ED), even though he knew it was useless.*
(P. 17)

Some patients perceived that the wards and waiting areas were overcrowded with patients, patients’ relatives, etc. As a result, they accepted the measures in place to limit the number of people accompanying them in the ED as they understood that this would reduce overcrowding and noise levels.


*It was like a fairground (the ED), patients on stretchers, on the floor, family members, everyone making noise, talking loudly, even laughing… It’s not an appropriate place for that, to be honest.*
(P. 13)

In the same way that loneliness, feelings of neglect, and lack of family or social support can threaten dignity, patients sometimes reported that receiving excessive or inappropriate visits from people who “should not be coming” could also cause discomfort, overcrowding, and a perceived loss of dignity.


*… Even those who were not supposed to come to see me have come to see me (while in the ED). And sometimes I’m not in the mood for so many visits, not even in the ER can you have so many visits.*
(P. 1)

Another key aspect of dignity mentioned by the participants was concern about posthumous events. These worries can be heightened in an emergency setting where patients may feel that they are not only dealing with their own health but also with the emotional burden of their loved ones. Fearing that loved ones will argue, be left alone, or lack financial stability can be a source of unease that leads the patients to feel vulnerable. For example, one patient was concerned that their relatives would argue and gave advice on how to resolve these disputes respectfully.


*I tell them, honey, don’t fight, among siblings, among cousins, respect each other a lot, and if someone has to compromise, you compromise.*
(P. 15)

Among the worries patients had about the period following their death, those of a financial nature were of particular concern. One patient informed us that he wanted to get married as soon as he learned of his advanced or terminal illness in order to guarantee a widow’s pension for his partner:


*I can’t leave (die) yet. In March I will marry Concha. On the 20th or 21st of March. So that she can get a better pension.*
(P. 6)

The participants were also concerned that their spouses would be lonely. They were comforted by the prospect of having someone to live with or to reconnect with. One participant spoke of his wife in the following way:


*She is still young… she has her mother who is a widow and lives alone. So, as her mother is alone, the two of them will be closer, they will be brought together again.*
(P. 8)

## 4. Discussion

The aim of this study was to explore the threats to dignity perceived by people with advanced illnesses who are treated in emergency departments. A qualitative approach allows for the research to be aligned with the patient’s perspective, thus adding social value to the research [18].

Two themes were developed from the data analysis that represent the main sources of threats to the dignity of people with advanced illnesses who are treated in EDs. The first is related to the characteristics of care in EDs and the complex situation of patients with advanced illnesses, which makes them more vulnerable. The close relationship between vulnerability and dignity at the end of life has been problematized in other studies [37], which stress that we must be mindful of the interplay between dignified vulnerability and vulnerable dignity [24]. This study also highlights the inextricable link between dignity in advanced illness and the experience of extreme existential and social vulnerability in EDs. Losing control over one’s decision-making process has been identified as a threat to dignity in this study. Similarly, another study highlighted the importance of supporting an individual in their decisions, even if we disagree [38], and the significance of feeling like a person and not a patient [39].

The persistence of professionals to establish a diagnosis, which is less relevant in the terminal stage of illness, has also been recognized as a source of undignified care in our study. Other studies have highlighted that patients may feel that their dignity is undermined when they are viewed as a “diagnosis” rather than as a unique individual [40]. Other threats to dignity identified by our participants included the priority given to treatment over comfort in triage systems. Attending the ED can in many cases involve long waits, mainly related to incorrect triage classification [41], as care for patients with advanced illnesses is often dismissed and seen as less of a priority by ED staff [42].

The patients who participated in this study highlighted the structural and organizational limitations of EDs as obstacles to their own dignity, which has already been recognized by healthcare professionals in other studies [28]. In EDs, it is difficult to guarantee the privacy of patients, who are cared for in an open-plan environment for the purpose of facilitating their observation and care [43]. This is compounded by the lack of privacy surrounding hygiene and elimination, which leads the patient to feel embarrassed and uncomfortable, while simultaneously compromising their dignity [44]. Minimizing the exposure of patients would require limiting the number of people accompanying the patient with advanced and/or terminal illness, which has proven to be useful [45]. Our participants complained of long waiting times, a lack of resources in the ED to cope with the high demand, and feeling helpless if they were not allowed to be accompanied by their loved ones. Consistent with these findings, a study in Indonesia related dignity to the speed of response when patients ask for something. It also highlights the importance of asking for consent before any intervention, having friendly conversations, and showing support [46].

The second theme of this study referred to the social dimension of dignity, which includes the people involved in patient care: professionals, other patients, family members, and loved ones who accompany patients in the ED. All of these groups shape or mold the experiences of dignity (and loss of dignity) in patients with advanced illnesses in EDs. Chochinov [27] included the social dignity inventory in his model of dignity preservation. This direct relationship between social conditions and dignity is evident in other studies [47]. In a recent review on dignity at the end of life, family and institutional dimensions were included in the dynamic reciprocity of dignity model [48].

Among the social actors who shape experiences of dignity (preservation or loss thereof), healthcare professionals stand out in particular [49,50]. The participants in this study perceived professionals, particularly nurses, to be great allies in preserving their dignity. Nevertheless, they also identified certain behaviors of professionals, including nurses, as genuine threats to dignity (lack of sensitivity, not listening to the patient, not respecting the patient’s autonomy, etc.). In a study carried out in Chile, nursing students also perceived this duality (guarantor and threat to dignity) in nursing professionals [51].

Within this social dimension, family members and loved ones can undermine the dignity of people with advanced illnesses in EDs. The participants of this study reported several threats related to dignity, such as feeling like they were a burden to others. Likewise, patients in another study stated that they would rather die than be a burden to their loved ones [39]. In line with studies involving professionals [28,52], the respective patients identified family obstinacy as a threat within the social dimension of dignity. Families often drive their loved ones with advanced illnesses to the ED because they are afraid or unable to care for themselves at home. Almost all participants in this study expressed concerns about the aftermath of their death. In line with other studies [53], these concerns related to leaving a will, the financial well-being of surviving family members, and the relationship between children. In contrast, individual spiritual concerns reported in other studies [54] were less pronounced in this study.

The main limitations of the study are related to the sample selection. Almost all of the participants’ advanced illnesses were cancer. Including other conditions could have enriched the results. All participants were adults, the majority were older adults (over 65 years), and two-thirds were men. Including younger people and a more balanced ratio of men and women might have yielded different results. Other limitations related to data collection are as follows: Participants could not be interviewed in the ED due to their clinical situation. Interviewing them at a later stage (sometimes days later) could have reduced the intensity of their memories. Moreover, it was sometimes difficult to confine the interview to their experience of treatment in the ED as patients digressed considerably. Only one interview per participant was conducted so that the experiences and perceptions shared would only refer to the most recent ED visit. A third set of limitations has to do with the rigor of the data analysis. Due to the clinical situation of the participants, the content of the transcripts or the analysis was not given to them for their approval or feedback. Finally, given the nature of qualitative research, it is not possible to generalize or extrapolate the results beyond the context of this study.

## 5. Conclusions

People with advanced or terminal illnesses who attend EDs may have their dignity undermined as the care they expect to receive in EDs does not align with the prevailing approach to care in EDs, which is focused on diagnosis and medical treatment. The structure and organization of EDs, together with the vulnerable situation of people with advanced illness, constitute a threat to patient dignity. There is an interrelation between social context and dignity, whereby health professionals, relatives, and loved ones can pose a threat to the patient’s dignity. This can either be due to how they are treated (obstinacy, lack of sensitivity) or to the self-imposed demands of the person with advanced illness (not being a burden, concerns regarding the aftermath of their death).

To address these perceived threats to dignity in patients with advanced illness, changes in protocols and the structure of EDs are required. For example, to limit overexposure and loss of privacy, the most appropriate location should be identified or specific spaces should be available to accommodate patients with advanced or terminal illness. Likewise, these patients should be asked if they wish to be accompanied, which should be facilitated where appropriate. To ensure a greater emphasis on the needs of the patients with advanced illnesses in EDs, protocols should be developed and professionals trained to provide care focused on comfort and dignity, rather than on diagnosis and treatment. Furthermore, patients with advanced illness should have available resources to ensure that they feel listened to, comforted, and respected when visiting EDs. Lastly, further research should be conducted on interventions that promote dignity in the ED.

## Figures and Tables

**Figure 1 healthcare-12-02581-f001:**
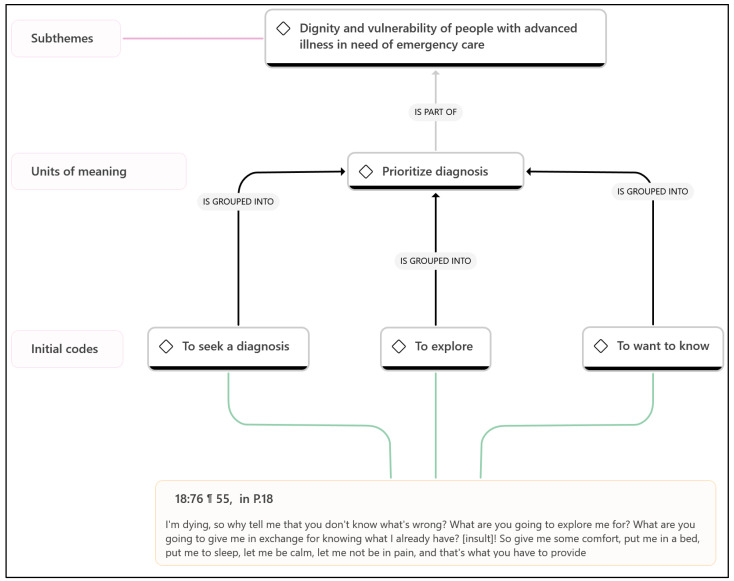
Example of the process of generating units of meaning and subthemes. 18:76: quotation number 76 in the document number 18. ¶ 55: paragraph. P18: Participant 18.

**Table 1 healthcare-12-02581-t001:** Sociodemographic data of the participants.

Interviews (n = 18)	% (n)
Age	
51–60	16.67 (3)
61–70	16.67 (3)
71–80	38.89 (7)
81–90	27.77 (5)
Sex	
Male	66.77 (12)
Female	33.33 (6)
Country of origin	
Spain	88.88 (16)
Ecuador	5.56 (1)
United Kingdom	5.56 (1)
Occupation	
Retired	72.22 (13)
Artist	5.56 (1)
Construction work	5.56 (1)
Farmer	11.11 (2)
Marble worker	5.56 (1)
Diagnosis	
Gastric cancer	22.22 (4)
Prostate cancer	5.56 (1)
Colon cancer	16.67 (1)
Bladder cancer	11.11 (2)
Lung cancer	22.22 (4)
Melanoma	5.56 (1)
Breast cancer	5.56 (1)
Liver cirrhosis	5.56 (1)

**Table 2 healthcare-12-02581-t002:** Interview script.

Stage	Topic	Content/Sample Questions
Introduction	Purpose	The belief is that their experience provides information that should be common knowledge.
	Objective	Conduct research to shed light on their experience.
Opening	General introductory question	If you agree, let us start with you telling me about your experience in the emergency department. Why did you go there?
Development	What does the word “Dignity” in the ED suggest to you? How do you perceive your dignity to be undermined when you go to the ED? How do you perceive the care received by professionals and how does it relate to dignity?	
Closing	Final question	Is there anything else you would like to add to the topic?
	Acknowledgments	We thank you for the time you have spent with us. We would like to remind you that your contribution will be of great help to us.

**Table 3 healthcare-12-02581-t003:** Themes, subthemes, and units of meaning.

Theme	Subtheme	Units of Meaning
Theme 1. When care focused on diagnosis and treatment limits the dignity of the patient with advanced illness	Dignity and vulnerability of people with advanced illnesses in need of emergency care	Loss of control Non-priority patient Prioritize treatment Prioritize diagnosis
	Structural and organizational constraints of EDs and their impact on dignity	High waiting time Lack of resources Feeling helpless Loss of privacy Limiting company
Theme 2. The social dimension of dignity in people with advanced illnesses in emergency departments	Healthcare professionals as both protectors and threats to dignity	Nurses as the backbone of the system Lack of sensitivity and warmth Changing perspective Listening to the patient’s story Justifying professionals
	How patients, family, and loved ones shape experiences of dignity preservation or loss	Feeling like a burden to relatives Family obstinacy Too many patient companions Inappropriate visits Posthumous concerns

## Data Availability

The data are held by the senior author (J.G.-M.) and are available to anyone interested. The data (transcripts and audio) contain information that may compromise the confidentiality and privacy of the participants (names of individuals, family members, and professionals) and therefore cannot be made public.

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
