# Peer review of "Threats to the Dignity of People with Advanced Illness Who Are Treated in Emergency Departments: A Qualitative Study"

_healthcare, 2024, doi:10.3390/healthcare12242581_

Round 1
Reviewer 1 Report
Comments and Suggestions for Authors
In overall this is a good topic.
Title- the full stop should be replaced with colon, ie ... department: a qualitative study.
Usually ethic approval is reported in Methods rather than abstract.
Line 54: last months- define (is it last 12months before demise or..?) This will help reader to understand without need to look at references.
Methods: Inclusion criteria. (1) advanced disease- noted in Table 1 almost all are malignancies. Are they early stage or advanced or terminal. Shouldn't advanced disease also include advanced kidney failure, heart failure, COPD D/E, etc? Why is it that only cancer is included?
Exclusion criteria line 83- (3) refusing to participate. this need not to be mentioned as this is given and understood.
Data collection. Interviews took place between 2020 to 2023. This is highly unusual and ethics approval was given in 2016. What happened that caused such a delay? in the span of three years how do you ensure that the interviewers were trained similarly? Any practicality issues apart from impact COVID-19 that was encountered by the investigators that resulted only 18 participants being interviewed?
Data saturation or thematic saturation was not described.
Results
66.67% and 33.33% doesn't make much sense for gender and age. Just round up to one decimal place. 71.83 too. STD should be abbreviated as SD.
Line 212. P-16 should be (P.16)
Quotes selected are apt in the first few sub themes.
Towards the end, the quotes tend to digress (this has been acknowledged in limitation part) and not focussing to ED. Can the team look at the last few sub themes and relate with dignity during presentation at ED?
Line 439. Due
Author Response
Comment: In overall this is a good topic.
Response: Thank you for your comment.
Comment: Title- the full stop should be replaced with colon, ie ... department: a qualitative study.
Response: Thank you for pointing this out. We have changed the title by replacing the full stop with a colon.
Usually ethic approval is reported in Methods rather than abstract.
Response. We agree. We have moved the sentence on ethical approval to sub-section 2.5 Ethical issues, in the Method section.
Line 54: last months- define (is it last 12months before demise or..?) This will help reader to understand without need to look at references.
Response. It is indicated that, according to the references cited (19), last months refers to the last 6 months of life.
Methods: Inclusion criteria. (1) advanced disease- noted in Table 1 almost all are malignancies. Are they early stage or advanced or terminal. Shouldn't advanced disease also include advanced kidney failure, heart failure, COPD D/E, etc? Why is it that only cancer is included?
Response. The patients are in advanced or terminal stages of their disease. Although most of them are cancer patients, there is also one patient who had liver cirrhosis. We have acknowledged this isolated case in the limitations. It could be that the selection of participants was biased or that there were very few visits to the emergency department for advanced diseases other than cancer during this period.
Limitations: Almost all of the participants' advanced illnesses were cancer. Including other conditions could have enriched the results.
Exclusion criteria line 83- (3) refusing to participate. this need not to be mentioned as this is given and understood.
Response. Thank you for your comment. The exclusion criterion has been removed (4)
Data collection. Interviews took place between 2020 to 2023. This is highly unusual and ethics approval was given in 2016. What happened that caused such a delay? in the span of three years how do you ensure that the interviewers were trained similarly? Any practicality issues apart from impact COVID-19 that was encountered by the investigators that resulted only 18 participants being interviewed?
Response. In 2016, the first version of the project was drafted so that we could apply for project funding. Permission was then obtained from the ethics committee. The funding was granted in 2017 and the project was implemented between 2018 and 2021, which was extended to 2022 due to COVID. In 2020, following the pandemic, the first interviews were held. In 2022, a researcher withdrew from her doctoral studies, citing personal and work-related reasons. This PhD Student was replaced by another researcher (AFF), also a member of the funded project team, who had received the same training and is currently working on her PhD linked to this project. This new researcher completed the data collection in 2023 with two additional interviews to saturate the data in some categories.
Data saturation or thematic saturation was not described.
Respuesta. We describe data saturation in the data collection sub-section).
After analysing the first interviews, they continued until no new themes emerged and the researchers considered that data saturation had been reached.
Results
66.67% and 33.33% doesn't make much sense for gender and age. Just round up to one decimal place. 71.83 too. STD should be abbreviated as SD.
Response. Thank you for your comment. The numbers have been corrected to one decimal place. The abbreviation for Standard Deviation (SD) has been corrected.
Line 212. P-16 should be (P.16)
Response. Thank you. The way in which the participants’ quotations are cited has been amended and standardized throughout the document. (P.1), (P.x)
Quotes selected are apt in the first few sub themes.
Towards the end, the quotes tend to digress (this has been acknowledged in limitation part) and not focussing to ED. Can the team look at the last few sub themes and relate with dignity during presentation at ED?
Response. Thank you. We have attempted this with the subtopics less related to ED.
Line 439. Due
Response. Thanks. Corrected. Now in line 444.
Reviewer 2 Report
Comments and Suggestions for Authors
Review
This is an interesting study on the threats to maintaining the dignity of patients with terminal illnesses during their visit to the emergency department. The authors have analyzed a very important but often forgotten aspect of patient care in the emergency department - maintaining the dignity of the patient. The introduction provides sufficient information on the background and objectives of the study. The methodology is sound and correct. The results are presented clearly.
Main improvements:
I would expect the authors to improve the Discussion section by providing more detailed conclusions resulting from their analysis. It would be beneficial for the readers - potentially doctors, nurses, paramedics, hospital managers and family members of patients - to read suggestions on how they should improve their care, organization, behavior to help maintain the dignity of patients with advanced illnesses in the emergency department. The precise conclusions and suggestions can be presented in a table divided into advice for doctors, nurses/paramedics, family members, people responsible for organizing work in the emergency department.
Minor improvements:
Line 83 – please change (3) into (4)
Table 3 – Some sentences end with a dot and some without. I suggest to remove dots from the table.
Line 379 – please change “thr” into ‘the’
Line 409 – thereof? Or rather “there of”
Line 439 - please write “Due” instead of “ue”
The study should be published but requires refinement in the discussion section.
Author Response
This is an interesting study on the threats to maintaining the dignity of patients with terminal illnesses during their visit to the emergency department. The authors have analyzed a very important but often forgotten aspect of patient care in the emergency department - maintaining the dignity of the patient. The introduction provides sufficient information on the background and objectives of the study. The methodology is sound and correct. The results are presented clearly.
Response 1. Thank you for your comment. We really appreciate your overall assessment of our work.
Main improvements:
I would expect the authors to improve the Discussion section by providing more detailed conclusions resulting from their analysis. It would be beneficial for the readers - potentially doctors, nurses, paramedics, hospital managers and family members of patients - to read suggestions on how they should improve their care, organization, behavior to help maintain the dignity of patients with advanced illnesses in the emergency department. The precise conclusions and suggestions can be presented in a table divided into advice for doctors, nurses/paramedics, family members, people responsible for organizing work in the emergency department.
Response 2. Thank you for the suggestion. We have included a new paragraph in the conclusions section with implications for practice, policy and research.
Implications for policy, practice and research. To limit overexposure and loss of privacy, one should identify the most appropriate location or have specific spaces available to accommodate patients with advanced or terminal illness. Likewise, these patients should be asked if they wish to be accompanied, which should be facilitated where appropriate. To ensure a greater emphasis on the needs of the patients with advanced illness in EDs, protocols should be developed and professionals trained to provide care focused on comfort and dignity, rather than diagnosis and treatment. Furthermore, patients with advanced illness should have available resources to ensure that they feel listened to, comforted and respected when visiting EDs. Lastly, further research should be conducted on interventions that promote dignity in the ED.
Minor improvements:
Line 83 – please change (3) into (4)
Response 3. Criterion 3 has been deleted at the request of another reviewer.
Table 3 – Some sentences end with a dot and some without. I suggest to remove dots from the table.
Response 4. Thank you for this recommendation. All full stops in table 3 have been deleted.
Line 379 – please change “thr” into ‘the’
Response 5. Thank you, this has been corrected.
Line 409 – thereof? Or rather “there of”
Response 6. A native English-speaking professional translator has confirmed that this is correct. The sentence has been re-worded.
Health care professionals in particular stand out among the social factors that shape experiences of dignity, whether preservation or loss thereof [49].
Line 439 - please write “Due” instead of “ue”
Response 7. Thank you, this has been corrected.
The study should be published but requires refinement in the discussion section.
Response. Thanks for your comment. We had refined the discussion section.
Reviewer 3 Report
Comments and Suggestions for Authors
This paper deals with an interesting topic -dignity of people with advanced illness- and develops qualitative research based on in-depth interviews.
However, in my opinion the article has several weaknesses:
1. The whole study deals with human dignity and threats to dignity perceived by people with advanced illnesses. But no clarification is provided on the concept of dignity used by the authors. As human dignity is the central topic of the paper, an explanation of what the authors understand by human dignity would be necessary. Moreover, the authors mention in line 38 that dignity is considered a human right. Only a minority of legal scholars would endorse this point of view, as for the majority dignity is not a specific right but the basis of all rights. It seems that authors confuse the concept of human dignity with that of human treatment. More accuracy in the use of legal terms should be required.
2. Concerning the in-depth interviews, their orientation is not impartial but seems to be directed to a certain outcome. As Table 2 (page 3) shows, one of the central questions of the interview is: “How do you perceive your dignity to be undermined when you go to the ED?”. This question is not formulated in a neutral manner. It clearly leads the person interviewed to do a negative assessment of his or her experience in ED, and therefore conditions the results of the research. Maybe with a neutral question the experiences described in the interviews should not have been so negative.
3. In my opinion, paragraph 4, which contains the discussion of the research results, is not very clear. On the one hand, there is a certain confusion among the conclusions provided by previous studies to which the authors refer and those of the present research. On the other hand, the discussion is not fully coherent with the results described in paragraph 3, as it introduces conclusions to which no reference was made in the previous paragraph. For example, in lines 376-377 it can be read that the study “has highlighted the importance of supporting an individual in their decisions, even if we disagree”, which was not mentioned before. The paragraph contains also some linguistical or typographical mistakes: in line 376, “This study has found the loss of control to be a threat to dignity in this study”, the words “this study” should not be repeated; in line 379, the word “thr”, it should be “the” or “their”. Finally, as a limitation of the research the authors do not mention that most of the participants are men (12 men to 6 women), which seems to me also important to evaluate the results.
4. Paragraph 5, which contains the conclusions, provides only a brief summary of the discussion (of paragraph 4). I think that some policy recommendations should be added, at least to provide some suggestions for future research. If people with advanced or terminal illness perceive that their dignity is undermined in Eds, at least some hints on what should be done to avoid this situation should be given.
Author Response
This paper deals with an interesting topic -dignity of people with advanced illness- and develops qualitative research based on in-depth interviews.
However, in my opinion the article has several weaknesses:
- The whole study deals with human dignity and threats to dignity perceived by people with advanced illnesses. But no clarification is provided on the concept of dignity used by the authors. As human dignity is the central topic of the paper, an explanation of what the authors understand by human dignity would be necessary. Moreover, the authors mention in line 38 that dignity is considered a human right. Only a minority of legal scholars would endorse this point of view, as for the majority dignity is not a specific right but the basis of all rights. It seems that authors confuse the concept of human dignity with that of human treatment. More accuracy in the use of legal terms should be required.
Response 1. Thank you for your comment. Although dignity is mentioned in Article 1 of the Declaration of Human Rights, the declaration does indeed take this concept (dignity) as the basis of all rights, as stated in the preamble. We have therefore corrected the sentence to reflect this. We also provide an explanation of dignity from the viewpoint of the German philosopher Immanuel Kant, whose perspective is the basis of other research we have conducted on dignity.
Dignity is at the core of the UN Universal Declaration of Human Rights [2]. From a Kantian perspective, dignity is rooted in the autonomy and free will of rational beings, who are capable of self-legislation and must treat themselves and others as ends in themselves, never merely as means to other ends. Dignity represents an intrinsic, immeasurable value derived from a person’s inner worth.
- Concerning the in-depth interviews, their orientation is not impartial but seems to be directed to a certain outcome. As Table 2 (page 3) shows, one of the central questions of the interview is: “How do you perceive your dignity to be undermined when you go to the ED?”. This question is not formulated in a neutral manner. It clearly leads the person interviewed to do a negative assessment of his or her experience in ED, and therefore conditions the results of the research. Maybe with a neutral question the experiences described in the interviews should not have been so negative.
Response 2. You are right to consider that questions can prompt certain answers and that some may not be neutral. This study is part of a larger project with several objectives. The objective of this particular study is ‘to explore the threats to dignity perceived by people with ad-vanced illness who are treated in emergency departments’. Implicit in the research question is the hypothesis that there are indeed situations in which patients will perceive threats to their dignity. This is why we have included a question aimed at identifying such threats.
- In my opinion, paragraph 4, which contains the discussion of the research results, is not very clear. On the one hand, there is a certain confusion among the conclusions provided by previous studies to which the authors refer and those of the present research. On the other hand, the discussion is not fully coherent with the results described in paragraph 3, as it introduces conclusions to which no reference was made in the previous paragraph. For example, in lines 376-377 it can be read that the study “has highlighted the importance of supporting an individual in their decisions, even if we disagree”, which was not mentioned before. The paragraph contains also some linguistical or typographical mistakes: in line 376, “This study has found the loss of control to be a threat to dignity in this study”, the words “this study” should not be repeated; in line 379, the word “thr”, it should be “the” or “their”. Finally, as a limitation of the research the authors do not mention that most of the participants are men (12
Response 3: The discussion has been improved by clarifying at several points whether the ideas or conclusions provided refer to our study or to other previous studies.
The following sentence has been changed: “It has highlighted the importance of supporting an individual in their decisions, even if we disagree.” Para enfatizar que esa idea no es de nuestro estudio, sino de otro. (In this line a study highlighted…). In ours, we allude to losing control (in decision making) as being related to this.
This error, which was on line 376 and now on line 513, has been deleted: “This study has found the loss of control to be a threat to dignity in this study)
Typographical errors have been corrected.
The following sentence has been included in the limitations sections:
The main limitations of the study are related to the sample selection. Almost all of the participants' advanced illnesses were cancer. Including other conditions could have enriched the results. All participants were adults, the majority were older adults (over 65 years) and two thirds were men. Including younger people and a more balanced ratio of men and women might have yielded different results.
- Paragraph 5, which contains the conclusions, provides only a brief summary of the discussion (of paragraph 4). I think that some policy recommendations should be added, at least to provide some suggestions for future research. If people with advanced or terminal illness perceive that their dignity is undermined in Eds, at least some hints on what should be done to avoid this situation should be given.
Response 4. Thank you for the suggestion. We have included a new paragraph in the conclusions with implications for practice, policy and research.
Implications for policy, practice and research. To limit overexposure and loss of privacy, one should identify the most appropriate location or have specific spaces available to accommodate patients with advanced or terminal illness. Likewise, these patients should be asked if they wish to be accompanied, which should be facilitated where appropriate. To ensure a greater emphasis on the needs of the patients with advanced illness in EDs, protocols should be developed and professionals trained to provide care focused on comfort and dignity, rather than diagnosis and treatment. Furthermore, patients with advanced illness should have available resources to ensure that they feel listened to, comforted and respected when visiting EDs. Lastly, further research should be conducted on interventions that promote dignity in the ED.
Reviewer 4 Report
Comments and Suggestions for Authors
Dear authors
Global appreciation
The article adresses a very relevant topic fundamentally because you decided to study the perceptions of people with terminal ilness who resorted to the hospital urgency/ emergency services about the care received, the surrounding social and organizational conditions and the respectives interferences with their dignity in a circunstance they were more vulnerable and fragile. The théme is not easy to approch, even more if we consider the set of more less positive feelings emerging in situations of suffering associated to the proximity of the end of live.
Overall, the article is clear and reveals yours sensenvity toward sick people who in terminal stages of live should be treated with a holistic prespective and that does not endanger their inalienable human conditions and dignity, namely whem attending health services, in this case emergency services.
In the theoretical scope, you present an adequate framework and mobilize appropriate and relevant references. As the struture, it is logical and contains the usual sections in the configuration of an scientific article. I suggest some revisions to be made in the presentation and references.
Specific comments and recommendations
Title - You suggest that there are threats to the dignity to the terminal ill patients when they seek hospital emergency services and that you pretend know and analyse their perspectives of them.
Abstract - objective and with reference to the main content and methods mobilized. It summarizes the main results and conclusions of the suty.
Keywords - I suggest including threats
Introduction - Well strutured. It contextualizes and delimits the issues under study. It justified the same. It encompasses the conceptual theoretical framework, substantiating it using credible and current sources. Your main objetive is clearly defined.
Materials and Methods - Yours options are duly justified. Given the sample of the participants in the study, they seem appropriate to me. The investigation protocol and the procedures followeed reveal your attention and rigor in safeguarding freedom of expression and the confort e the dignity of the intervieweers. There are very positive points.
Please revise in the point 2.2 - lines 82 and 83 the numbering 3.
Results - After the presentation of the results, doubts remains about their representativiness and robustness. Perhaps it is possible a little more the way in wich you categorized them and defined the units of signification.
Please standardized the reference to the codes that represents the interviewed participants - e.g. (P1) and following.
Discussion - you mobilize the conceptual theoretical framework previously presented and enrich it with references to some more studies and authors. The discussion arouses interest and calls for reflection on the results obtained. You describe some limitations of the study including those related to the characteristics of the sample and the non validation to the participants themselves of the contents obtained from the interviews and respective treatment and discussion, wich is understandable and positive. You should however add the impossibility of generalysing and extrapolating the results and conclusions.
The conclusions are congruent with the results and answers to the main objective of the study. However i suggest that you presents some suggestions giving greater usefulness and interest to the study. In many other regions and countries there must be situations that would warrent shared reflection and further research.
References - were commented on throughout this review. I just suggest some standardization in their format.
Best regards.
Author Response
The article adresses a very relevant topic fundamentally because you decided to study the perceptions of people with terminal ilness who resorted to the hospital urgency/ emergency services about the care received, the surrounding social and organizational conditions and the respectives interferences with their dignity in a circunstance they were more vulnerable and fragile. The théme is not easy to approch, even more if we consider the set of more less positive feelings emerging in situations of suffering associated to the proximity of the end of live.
Overall, the article is clear and reveals yours sensenvity toward sick people who in terminal stages of live should be treated with a holistic prespective and that does not endanger their inalienable human conditions and dignity, namely whem attending health services, in this case emergency services.
In the theoretical scope, you present an adequate framework and mobilize appropriate and relevant references. As the struture, it is logical and contains the usual sections in the configuration of an scientific article. I suggest some revisions to be made in the presentation and references.
Response 1: We sincerely thank the reviewer for their positive comments. It is gratifying for us that you found the study interesting and well-conducted.
Title - You suggest that there are threats to the dignity to the terminal ill patients when they seek hospital emergency services and that you pretend know and analyse their perspectives of them.
Response 2: Thank you for your comment. We agree.
Abstract - objective and with reference to the main content and methods mobilized. It summarizes the main results and conclusions of the suty.
Response 3: Thank you. It has been slightly modified at the request of another reviewer to remove a sentence about the ethics committee approval. Substantially unchanged.
Keywords - I suggest including threats
Response 4: Thanks for the suggestion. We have included the keyword ‘threats’.
Introduction - Well strutured. It contextualizes and delimits the issues under study. It justified the same. It encompasses the conceptual theoretical framework, substantiating it using credible and current sources. Your main objetive is clearly defined.
Response 5: Thank you for your comment. We have added some definitions and clarified some terms.
Materials and Methods - Yours options are duly justified. Given the sample of the participants in the study, they seem appropriate to me. The investigation protocol and the procedures followeed reveal your attention and rigor in safeguarding freedom of expression and the confort e the dignity of the intervieweers. There are very positive points.
Response 6: Thank you for your kind comment.
Please revise in the point 2.2 - lines 82 and 83 the numbering 3.
Response 7: The exclusion criterion renumbered 3 has been eliminated.
Results - After the presentation of the results, doubts remains about their representativiness and robustness. Perhaps it is possible a little more the way in wich you categorized them and defined the units of signification.
Response 8: We expand on how we define units of meaning and themes/sub-themes, with a visual example using ATLAS.ti.
The results have been strengthened by focusing more on their relationship with EDs in some topics.
Please standardized the reference to the codes that represents the interviewed participants - e.g. (P1) and following.
Response 9. Thank you for your comment. The way of coding the participants’ quotations has been standardized. (P.1) (P.x)…
Discussion - you mobilize the conceptual theoretical framework previously presented and enrich it with references to some more studies and authors. The discussion arouses interest and calls for reflection on the results obtained. You describe some limitations of the study including those related to the characteristics of the sample and the non validation to the participants themselves of the contents obtained from the interviews and respective treatment and discussion, wich is understandable and positive. You should however add the impossibility of generalysing and extrapolating the results and conclusions.
Response 10: Thank you for your positive feedback. This following limitation has been added:
Finally, given the nature of qualitative research, it is not possible to generalize or extrapolate the results beyond the context of this study.
The conclusions are congruent with the results and answers to the main objective of the study. However i suggest that you presents some suggestions giving greater usefulness and interest to the study. In many other regions and countries there must be situations that would warrent shared reflection and further research.
Response 11. Thank you for the suggestion. We have included a new paragraph in the conclusions with implications for practice, policy and research.
Implications for policy, practice and research. To limit overexposure and loss of privacy, one should identify the most appropriate location or have specific spaces available to accommodate patients with advanced or terminal illness. Likewise, these patients should be asked if they wish to be accompanied, which should be facilitated where appropriate. To ensure a greater emphasis on the needs of the patients with advanced illness in EDs, protocols should be developed and professionals trained to provide care focused on comfort and dignity, rather than diagnosis and treatment. Furthermore, patients with advanced illness should have available resources to ensure that they feel listened to, comforted and respected when visiting EDs. Lastly, further research should be conducted on interventions that promote dignity in the ED.
References - were commented on throughout this review. I just suggest some standardization in their format.
Response 12: Thanks for the suggestion. We have standardized the format of the references.
Round 2
Reviewer 1 Report
Comments and Suggestions for Authors
The revised manuscript has demonstrated improvement and queries has been addressed. Added points support and help convey the idea better.
Just some typesetting
-pg6 Line 172 Eds--> EDs
-page 9 Line 298 ER--> ED (for standardisation)
-Page13 Eds --> EDs
Author Response
The revised manuscript has demonstrated improvement and queries has been addressed. Added points support and help convey the idea better.
Response. Thank you very much for your comment and for your suggestions to correct the remaining typesetting errors
Just some typesetting
- -pg6 Line 172 Eds--> EDs
Response 1. Thank you. This has been corrected (now on line 170)
- -page 9 Line 298 ER--> ED (for standardisation)
Response 2: Thank you for pointing this out. We have replaced ER with ED (now on line 295).
- -Page13 Eds --> EDs
Response 3. Corrected. Thank you for pointing out these errors.
Reviewer 2 Report
Comments and Suggestions for Authors
I am satisfied with the refinements.
The manuscript should be published.
Author Response
Coment 1. I am satisfied with the refinements. The manuscript should be published
Response 1. We are very grateful for your positive review and recommendation for publication.
Reviewer 3 Report
Comments and Suggestions for Authors
I think the authors have done an important effort to improve the quality of the paper. All the concerns I expressed in my first review have been met, except the fact that the in-depth interviews are not neutral, but authors explain in their response the reasons of this fact, which seem to me acceptable. I have only three very small formal suggestions:
a) Section 4 (Discussion) is still a little bit dificult to understand, because its mixes continuously conclusions of the present study with contributions of precedent studies. Although now the authors clearly identify which one are the conclusions of the present research, maybe if the section is divided in more paragraphs it would be more easy to read. For example, I would introduce a new paragraph in line 410, starting with the words " The persistence of professionals to establish a diagnosis...", and as well in line 441, starting with the words "Among the social actors..."
b) I suggest to change the first sentence of the last paragraph of the paper. Instead of "Implications for policy, practice and research", it would be better to introduce this paragraph with an explanatory sentence, such as: "To attend these perceived threats to dignity of patients with advanced illness, changes in protocols and structure of ED are required. For example..."
c) Although English is generally fine, I think it could still be improved, and I suggest to do a general revision.
These suggestions are intended only to continue improving the paper, but I appreciate the changes made and the effort of the authors to follow the recomendations of my first review.
Author Response
I think the authors have done an important effort to improve the quality of the paper. All the concerns I expressed in my first review have been met, except the fact that the in-depth interviews are not neutral, but authors explain in their response the reasons of this fact, which seem to me acceptable. I have only three very small formal suggestions:
a) Section 4 (Discussion) is still a little bit dificult to understand, because its mixes continuously conclusions of the present study with contributions of precedent studies. Although now the authors clearly identify which one are the conclusions of the present research, maybe if the section is divided in more paragraphs it would be more easy to read. For example, I would introduce a new paragraph in line 410, starting with the words " The persistence of professionals to establish a diagnosis...", and as well in line 441, starting with the words "Among the social actors..."
Response a. Thank you for your suggestion. We have added new paragraphs in the places suggested and we believe that this will indeed improve the readability of the document.
b) I suggest to change the first sentence of the last paragraph of the paper. Instead of "Implications for policy, practice and research", it would be better to introduce this paragraph with an explanatory sentence, such as: "To attend these perceived threats to dignity of patients with advanced illness, changes in protocols and structure of ED are required. For example..."
Response b. The first sentence of the second paragraph of the conclusions has been changed to improve the coherence of this section.
c) Although English is generally fine, I think it could still be improved, and I suggest to do a general revision.
Response c. The English has been proofread by a native British scientific text editor. We have attached a proofreading certificate.
These suggestions are intended only to continue improving the paper, but I appreciate the changes made and the effort of the authors to follow the recomendations of my first review.
Response. Many thanks. We believe that the reviewers' suggestions have undoubtedly contributed to improving our article.